# The effect of cold ischemia time on hypoxia, EMT, and apoptosis pathways in normal colon mucosa

Katarzyna Duzowska[1], Mikołaj Opiełka[1,2], Kinga Drężek-Chyła[1], Anna Kostecka[1], Monika Horbacz[1], Jarosław Skokowski[3], Olga Rostkowska[4], Jarosław Kobiela[4], Leszek Kalinowski[5,6], Jan P. Dumanski[1,7], Arkadiusz Piotrowski[1], Marcin Jąkalski[1,8☯*], Natalia Filipowicz[1☯*]

1 3P-Medicine Laboratory, Medical University of Gdańsk, Gdańsk, Poland, 2 Department of Biochemistry, Medical University of Gdańsk, Gdańsk, Poland, 3 Academy of Applied Medical and Social Science, Elbląg, Poland, 4 Department of Oncological, Transplant and General Surgery, Medical University of Gdańsk, Gdańsk, Poland, 5 Department of Medical Laboratory Diagnostics, Medical University of Gdańsk, Gdańsk, Poland, 6 BioTechMed Center, Department of Mechanics of Materials and Structures, Gdańsk University of Technology, Gdańsk, Poland, 7 Department of Immunology, Genetics and Pathology and Science for Life Laboratory, Uppsala University, Uppsala, Sweden, 8 Center for Applied Genomics and Bioinformatics, Faculty of Biology, University of Gdańsk, Gdańsk, Poland

☯ These authors contributed equally to this work
* marcin.jakalski@ug.edu.pl (MJ); natalia.filipowicz@gumed.edu.pl (NF)

## Abstract

Cold ischemia time (CIT), the interval between tissue excision and preservation, is a critical preanalytical variable that profoundly impacts gene expression profiles. Variability in CIT can lead to inconsistent transcriptomic results, making study interpretation challenging and undermining reproducibility in biomedical research. Our study aimed to evaluate the impact of CIT on the expression of cancer-related genes, particularly these involved in hypoxia, apoptosis, and epithelial-to-mesenchymal transition (EMT). We performed RNA sequencing on 54 normal colon mucosa samples from nine patients undergoing colorectal cancer surgeries, freezing samples at predefined intervals ranging from 0 to 60 minutes. A total of 44 differentially expressed genes (DEGs) ($p < 0.05$) were identified when comparing samples frozen immediately (T0) with those frozen after 60 minutes (T5). These DEGs were further analyzed through functional and pathway enrichment analyses and weighted gene co-expression network analysis (WGCNA). The enrichment analysis revealed significant alterations in pathways associated with apoptosis, hypoxia, EMT, and cancer progression, including p53 and HIF-1 signaling. WGCNA highlighted two co-expressed gene modules: ME2, which showed downregulation of apoptosis-related genes, and ME4, linked to apoptosis and cellular metabolism. Our findings highlight CIT as a critical preanalytical variable, showing that prolonged ischemia can induce transcriptomic changes that may mimic malignancy, and potentially confound research outcomes. To minimize such effects, we recommend keeping CIT under 30 minutes.

**Data availability statement:** All relevant data for this study are publicly available from the European Genome-phenome Archive (EGA) (https://ega-archive.org/datasets/EGAD00001015661).

**Funding:** This study was supported by the Foundation for Polish Science (FNP) under the International Research Agendas Program to J.P.D. and A.P., co-financed by the European Union under the European Regional Development Fund (MAB/2018/6); and the Ministry of Science and Higher Education, (grant No. 2/566516/ SPUB/SP/2023) to L.K.

**Competing interests:** The authors have declared that no competing interests exist.

## Introduction

The quality and integrity of biological samples are crucial for multi-omic studies, directly impacting the reliability of results. Preanalytical variables, including patient-specific factors and tissue handling protocols, can significantly impact the outcomes [1,2]. With biobanking playing a crucial role in biomedical translational research, standardized tissue collection protocols are essential to minimize external variability [3–5]. Ischemia, defined as a restriction of blood supply to tissues or organs, results in a shortage of oxygen supply necessary to sustain the cellular metabolism. In the *ex vivo* context, this process can be classified into warm and cold ischemia. Cold ischemia time (CIT) refers to the condition in which oxygen supply is entirely absent after tissue excision while the sample is maintained at temperatures below body temperature [6]. Warm ischemia time (WIT) refers to the period during which the tissue remains in the donor's body but the oxygen supply is insufficient to meet metabolic demands. This preanalytical variable is challenging to control as it relies on the efficiency of the surgical team, for which the priority is the successful completion of the operation. Reports on the impact of WIT on transcriptomic profiles are scarce, but studies, including those by Pedersen *et al*. and Ma *et al*., highlight its effects, identifying numerous differentially expressed genes (DEGs) and alterations in oncogenic, inflammatory, and immunological pathways [7,8]. In contrast, CIT is determined by the biobanking procedure and can be effectively managed by the tissue collector [9]. Prolonged CIT, rather than malignancy, can lead to specific, even subtle, changes in gene expression profiles, complicating the interpretation of tissue samples.

Several studies have assessed the impact of CIT on RNA quality, measured as RNA integrity number (RIN), reporting minimal changes up to 4 hours [10–12] and even up to 16 hours at room temperature [13]. RNA quality also varies by tissue type, with thyroid and colorectal tissues being more sensitive to cold ischemia compared to stomach and lung tissues [14]. Gastrointestinal samples, in general, tend to exhibit lower RNA quality than those from other organs [2,15], with tumor tissues displaying significantly higher RINs than normal tissues, though CIT has little effect on this observed difference [2].

While most reports focus on RNA integrity, few have explored the impact of CIT on gene expression, particularly in cancer-related pathways [11,16,17]. Transcriptomic analyses suggest that CIT exceeding 60 minutes can significantly alter gene expression. Aktas *et al*. identified 41 transcripts affected by prolonged CIT, including apoptosis- and cell cycle-related genes, with over 3% showing significant changes [10]. Other study reported transcriptomic alterations as early as 30 minutes post-excision, for instance delayed freezing of colorectal carcinoma samples lead to increased *KLF6* expression, potentially misrepresenting its role in tumor pathogenesis [16]. A study of transcriptomic profiles in renal carcinoma showed over 4,000 genes were affected at longer CIT durations and higher temperatures, emphasizing the need for immediate snap-freezing to preserve gene expression profiles relevant to cancer research [11].

Despite these findings, the existing literature remains limited, particularly regarding the effect of cold ischemia time (CIT) on the transcription of cancer-associated

genes, including those involved in epithelial-to-mesenchymal transition (EMT), hypoxia, apoptosis, and cell death. These genes are crucial for understanding cancer biology and the complex interactions within the tumor microenvironment [18]. Moreover, their expression can directly influence treatment decisions and patient outcomes [16,18–20]. Some of these genes may be first responders to cold ischemia, making them potential markers of CIT. Examining these preanalytical factors is essential not only to enhance the reliability and reproducibility of experimental findings but also to ensure that scientific conclusions translate effectively into clinical practice.

The primary aim of this research is to investigate the influence of CIT on the transcription profile of macro- and microscopically normal colon mucosa collected from patients with colorectal cancer. The focus on normal colon mucosa aligns with the design of ongoing research in our laboratory, which investigates genetic mosaicism focusing on post-zygotic mutations accumulating in morphologically normal tissues proximal and distal to tumors [4,21]. Furthermore, the majority of the publicly available studies focused on tumor tissue, with limited insights into healthy tissue [2,13,17]. Here we employed targeted RNA sequencing (NGS RNAseq), including, among the others, genes associated with hypoxia, apoptosis, and cancer, thus addressing a critical gap in the literature on preanalytical variables for tissue collection and subsequent analysis for cancer research.

## Materials and methods

### Study design and Sample Handling

The healthy colon mucosa samples were collected from 9 consecutive patients (2 females; 7 males; mean age 68 years ± 10.9) diagnosed with colorectal cancer who underwent surgical resection from 21st of April to 24th of August 2021 in the Department of Oncological, Transplant, and General Surgery; University Clinical Center in Gdansk. The summary of patients' data is presented in Table 1. All the procedures were performed using an open surgical approach, which minimizes WIT by allowing immediate specimen extraction following devascularization. In contrast, laparoscopic and robotic approaches require placement of the devascularized specimen into an extraction bag, with removal occurring only at the completion of the procedure. All operations consisted of left-sided colectomies or rectal resections, with operative times ranging from 1 hour 15 minutes to 3 hours 20 minutes. Considering the open approach and the review specific technical aspects of the procedures, WIT did not exceed 40 minutes in any case. This variable should be considered unmodifiable, as it is largely determined by intra-abdominal anatomical conditions. After the surgical resection, a larger, full-walled specimen of the colon was immediately excised approximately 10–15 cm away from the primary tumor, rinsed in saline and left at room temperature until the mucosa collection procedure at a particular time point was completed. At each time point, a sample of macroscopically unaffected mucosa was excised by dissecting from the colon specimen and detaching gently from the muscular layer. Subsequent washing steps were implemented, including: saline solution (twice), antibiotic

**Table 1. A summary of donors and diagnoses included in the study.**

| Patient ID | Sex | Age | ICD-10 Classification |
|---|---|---|---|
| ES31C | Male | 67 | C18.7 - Malignant neoplasm of sigmoid colon |
| 5IDSR | Female | 70 | C19 - Malignant neoplasm of rectosigmoid junction |
| OV1IW | Male | 67 | C20 - Malignant neoplasm of rectum |
| NYYCH | Male | 48 | C20 - Malignant neoplasm of rectum |
| LZZOP | Female | 72 | C19 - Malignant neoplasm of rectosigmoid junction |
| JSDHP | Male | 68 | C19 - Malignant neoplasm of rectosigmoid junction |
| PZ94B | Male | 83 | C18.7 - Malignant neoplasm of sigmoid colon |
| FFMB1 | Male | 59 | C20 - Malignant neoplasm of rectum |
| ABCD | Male | 83 | C18.5 - Malignant neoplasm of splenic flexure |

solution (Penicillin – Streptomycin 5000 U/ml), and saline solution, followed by snap freezing in liquid nitrogen. Tissue fragments were collected at six time points – 0 (T0), 10 (T1), 20 (T2), 30 (T3), 45 (T4), and 60 (T5) minutes after the surgical resection, estimated as the time of cold ischemia. Sample collected at T0 was processed immediately after organ resection. First time point (T0) was used as the reference and the starting point for all the subsequent analyses. In total 54 samples were collected and subjected to further analysis.

## RNA extraction and quality control

Colon mucosa samples were stored at −80°C until RNA isolation. Total RNA was extracted from 10–30 mg of tissue samples that were mechanically homogenized using T10 Basic ULTRA – TURRAX disperser (IKA) in the presence of QIAzol Lysis reagent (Qiagen). RNA isolation, purification, and DNase digestion were performed using the RNeasy Mini kit (Qiagen) according to the original protocol with two modifications as described in the previous paper [22] RNA quality and quantity were assessed using TapeStation (Agilent Technologies) using RNA ScreenTape kit according to the manufacturer's protocol (S1 Table in S1 File). RNA integrity number (RIN) was calculated using TapeStation analysis software (Agilent Technologies). Additionally, the DV200 index, indicating the percentage of RNA fragments > 200 nt, was calculated and used as a quality assessment standard [23]. Samples with a DV200 index > 75% were used for further NGS analysis.

## Targeted RNA sequencing

The targeted RNA sequencing panel was designed with the Roche NimbleDesign online tool (Roche, now Hyper-Design, https://hyperdesign.com/#/) and covered 7229 regions with a total length of 1,243,523 bp. It included 634 genes based on literature research and covered genes associated with apoptosis, cell death, hypoxia, epithelial-to-mesenchymal transition, and other cancer-related genes (the full list of transcripts is given in S2 Table in S1 File). NGS libraries were prepared with the Kapa RNA HyperPrep Kit (Roche) using Automated Liquid Handling Bravo NGS workstation (Agilent Technologies) according to the manufacturer manual with 100 ng RNA used as an input and enzymatic fragmentation at 94°C for 6 minutes, with the addition of ERCC RNA Spike-In Mix (Invitrogen) as an external RNA control. All the single libraries were multiplexed and hybridized with SeqCap EZ Choice Probes (Roche) designed by our group and KAPA HyperCapture Reagent and Bead kit (v2, Roche Sequencing Solutions, Inc.) according to SeqCap RNA Enrichment System User's Guide (v.1.0) with slight modifications. Component A was replaced with formaldehyde, and the Multiplex Hybridization Enhancing Oligo Pool was replaced with Universal Blocking Oligos (UBO). The hybridization was run for 18h, the library was cleaned up, post-captured amplified, and purified, followed by inspection of fragment distribution using TapeStation with High Sensitivity D1000 Screen Tape kit (Agilent Technologies) and qPCR quantitation in a Roche LightCycler 480 with KAPA Library Quantification Kit (Roche). Paired-end reads of 150 bp were generated using TruSeq RNA Access sequencing chemistry by an external service provider (Macrogen Europe, Amsterdam, The Netherlands).

## Transcriptomic data analysis

The RNA-seq data were processed as described in Andreou et al. 2024 [21]. Briefly, after the quality filtering and trimming of the raw FASTQ files with BBDuk (https://sourceforge.net/projects/bbmap/, version 38.36), the resulting reads were mapped to the reference human genome (hg38, GENCODE version 39) using STAR version 2.7.3a [24]. Raw read counts assigned to the annotated genes obtained in the above process were collated into a single gene expression matrix and processed further in R programming language (https://www.r-project.org, version 4.1.2). Lowly expressed genes were filtered out based on a minimal required CPM expression threshold. The filtered gene expression matrix was normalized using the TMM method in edgeR [25]. Principal Component Analysis (PCA) was performed to investigate sample grouping and identify potential outliers using FactoMineR, version 2.4 [26]. We accounted for the paired nature of the RNAseq data

by treating the patient ID as batch and removing its effect from the data using ComBat-seq [27]. Significantly differentially expressed genes (DEGs) were identified with EdgeR using the glmLRT function (likelihood ratio test) with a significance threshold set to 0.05 (False Discovery Rate, FDR).

Weighted gene co-expression network analysis (WGCNA) was performed on the normalized and batch-adjusted gene expression values using the WGCNA R library WGCNA [28]. Soft threshold value for building the correlation network was selected empirically based on diagnostic plots (pickSoftThreshold function). The final network was built using the blockwiseModules function, where the TOMType parameter was set to "signed". Subsequently, a linear model (lmFit, from the limma package [29]) was run on all modules to identify those associated with the tested trait (time point).

### Overrepresented terms/ GO/ pathways

Overrepresentation analysis (ORA) was used to perform functional enrichment analysis of significantly differentially expressed genes, utilizing the Gene Ontology (GO) database and the genome reference set in WebGestalt [30]. Pathway enrichment analysis was similarly conducted with ORA, focusing on KEGG pathways and the genome reference set in WebGestalt. In addition, functional and pathway enrichment analyses were separately applied to genes from WGCNA modules ME2 and ME4, using WebGestalt for the analysis.

### Bioethics committee approval

All procedures for sample collection were approved by the Independent Bioethics Committee for Research at the Medical University of Gdansk (approval number NKBBN/564/2018 with multiple amendments). Written informed consent was obtained from all the patients prior to surgery. All procedures were performed in accordance with the relevant national and international laws and guidelines as well as in compliance with European Union General Data Protection Regulation (EU GDPR).

## Results

### Quality of total RNA within the studied time frame

Tissues from nine patients were collected and snap-frozen at six time points: immediately post-resection (T0), 10 minutes (T1), 20 minutes (T2), 30 minutes (T3), 45 minutes (T4), and 60 minutes (T5) post-resection (see Materials and Methods). As many prior studies have focused on RNA quality as a consequence of CIT, we performed quality control by measuring both RIN and DV200 values. Notably, RIN values remained consistent across all analyzed time points, with a median value of 4.3. Furthermore, all samples had DV200 above 75%, indicating that RNA was of sufficient quality for subsequent analysis [23]. The consistency of these results across all time points suggest that RNA integrity was preserved throughout the collection and processing protocol. Detailed RIN measurements for each sample are provided in S1 Table in S1 File.

### Differential expression analysis highlights genes associated with hypoxia, EMT, and apoptosis

Differential expression analysis revealed 44 significantly differentially expressed genes (DEGs) between the two extreme time points set for our study, T0 and T5 (60 min) (S3 Table in S1 File). Principal component analysis (PCA) demonstrated a noticeable separation between samples at T0 and T5 after eliminating patient batch effect, as illustrated in Fig 1. Comparisons involving T0 and other time points (T1–T4) showed fewer statistically significant DEGs: zero between T0 vs T3, and six for T0 vs T4, namely *ABCB1*, *ABCG2*, *EPAS1*, *FZD2*, *LAMA3*, and *TMPRSS2*. Our results show that the initial gene expression changes occur between 30 minutes (T3) and 45 minutes (T4) post-excision; with 44 genes displaying significant changes between 45 minutes and 60 minutes (T5) after excision (S3 Table in S1 File). Thus, our further analyses focused on the T0 versus T5 comparison.

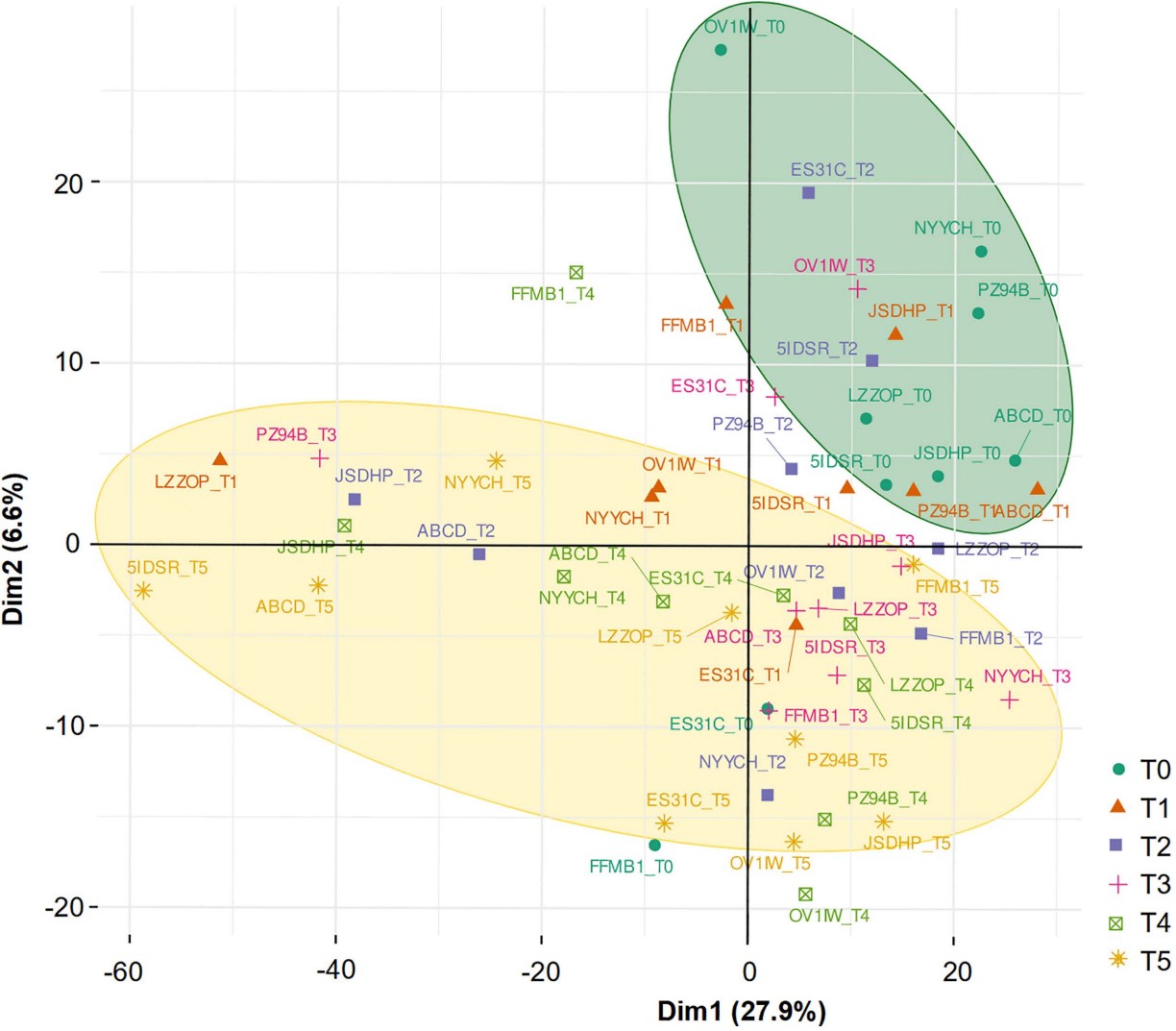

**Fig 1. Principal component analysis (PCA) of all 54 samples using the normalized batch-corrected expression profiles of 1419 genes (both targeted and non-targeted).** Each point represents the orientation of a sample projected into the transcriptional space, color and symbol refer to its group membership. The analysis demonstrated clearly separated clusters of T0 and T5 samples (marked in green and yellow circles), a pattern not pronounced among the intermediate time points (T1-T4). T0 - time point 0, frozen immediately after resection; T1 - time point 1, frozen 10 minutes after resection; T2 - time point 2, frozen 20 minutes after resection; T3 - time point 3, frozen 30 minutes after resection; T4 - time point 4, frozen 45 minutes after resection; T5 - time point 5, frozen 60 minutes after resection.

Applying an absolute logFC cutoff of ≥1, we identified 15 upregulated genes and 29 downregulated genes in T5 compared to T0. Notably, DEGs included hypoxia- and EMT-associated genes such as *SNAI, CDH1, ADM, MMP9, EGLN3, IL-6, AKT3, TSPAN1*, and *DSC2*, as well as apoptosis-related genes: *CASP7, TNFSF10, DEFB1, AKT3*, and *JPH3*. Among the 44 DEGs identified between T0 and T5 (S3 Table), 5 were linked to hypoxia, 8 to apoptosis or cell death, and 10 to epithelial-to-mesenchymal transition (EMT) determined using EMTome [31]. The remaining 19 were classified as 'other'. It is important to stress that none of 22 housekeeping genes used as a reference set included in the panel showed significant expression changes. S3 Table in S1 File provides the complete DEG list with the logFC, p-values, and FDR values.

Functional enrichment of DEGs using overrepresentation analysis (ORA) identified enrichment of Gene Ontology (GO) terms associated with processes such as apoptosis, cell-cell adhesion, tissue migration, and others. TNF signaling, ABC transporters, gastric cancer, and cancer pathways were among the enriched pathways shown by ORA KEGG analysis (S4 Table in S1 File.).

## Weighted Gene Co-Expression Network Analysis (WGCNA) confirms two gene modules associated with apoptosis, EMT and hypoxia

To find the clusters of highly correlated genes in our dataset, we applied the WGCNA analysis that identified two significant gene modules, ME2 and ME4 (adjusted p-values 0.037 and 0.018, respectively), which demonstrate distinct patterns during the time course of our study. Notably, the genes within these modules were associated with key biological processes, including apoptosis, EMT, and hypoxia-related pathways (Fig 2).

ME2 consisted of 145 genes, primarily exhibiting a downregulation pattern in T5 in relation to T0 (Fig 2A, 2B). Among these, 26.2% were related to apoptosis or cell death, 18.6% to EMT, 11.7% to hypoxia, and 0.69% (one gene) was annotated as a reference gene in our targeted panel (Fig 2C). The remaining 42.8% were categorized as 'other' (see details in S5 Table in S1 File). Functional enrichment analysis of ME2 genes using ORA GO (FDR < 0.05) identified several pathways, including those related to the execution phase of apoptosis and the extrinsic apoptotic signaling pathway. KEGG pathway analysis further highlighted nine pathways, including cancer-related and apoptosis-related processes such as p53 signaling, HIF-1 signaling, and FoxO signaling (S7 Table in S1 File).

ME4, a smaller gene module consisting of 56 genes, also displayed a downregulation pattern (Fig 2A, 2B). Genes comprised in this module showed a faster response to CIT, reaching a stable point at the T4 time point (Fig 2A). In ME2, however, the difference between T4 and T5 is visibly larger (Fig 2A). Within the ME4 cluster, 25% of genes were identified as apoptosis-related, while 16.1% were associated with EMT and hypoxia, respectively. Furthermore, 8.93% were classified as reference genes and 33.9% as 'other' (Fig 2A, 2B). ORA GO analysis (FDR < 0.05) in ME4 identified pathways connected to numerous metabolic processes, cell-cell signaling and homeostasis. Similarly, KEGG pathway analysis revealed significant metabolic and cancer-related processes, including glycolysis/gluconeogenesis, HIF-1 signaling pathway, and pathways related to gastric cancer (S7 Table in S1 File.).

## Discussion

We examined the impact of up to 60 minutes of CIT on the transcription of cancer-related genes, with samples snap-frozen at defined time points. RNA integrity (RIN) varied across samples without a clear correlation to CIT (S1 Table in S1 File), consistent with studies showing minimal RNA degradation after 3–4 hours [12,13,32,33]. Despite applying four rinsing steps, colon mucosa samples showed relatively low RNA quality (mean RIN = 4.4, range 2.7–7.7), aligning with findings from other tissue comparisons [14,15]. This may stem from enzymatic activity, bacterial influence, or tissue compartment differences [34]. To mitigate this, RNA fragmentation conditions were optimized during the process of NGS library preparation.

Although utilizing a comprehensive transcriptomic approach covering over 600 genes, we detected no DEGs within the first 30 minutes post-resection, indicating this timeframe is likely safe for tissue excision and snap-freezing. Only six DEGs were identified between the 0 and 45 minute time points (T4), while 44 DEGs emerged between the 0 and 60 minute time points (T5), highlighting this period as critical for tissue freezing. Notably, five of the six DEGs in T4 and 29 of the 44 DEGs in T5 were downregulated. These results align with findings from Aktas *et al.*, who reported 41 transcripts affected by prolonged CIT, including genes involved in cell cycle regulation, apoptosis, stress responses, and cancer progression [10]. Although this number seems modest, it might have significant implications for researchers focusing on specific genes, gene signatures, or pathways, particularly because these DEGs are closely linked to cancer development and progression. The broader impact of sample handling on the transcriptome varies widely across studies. von der Heyde *et al.*

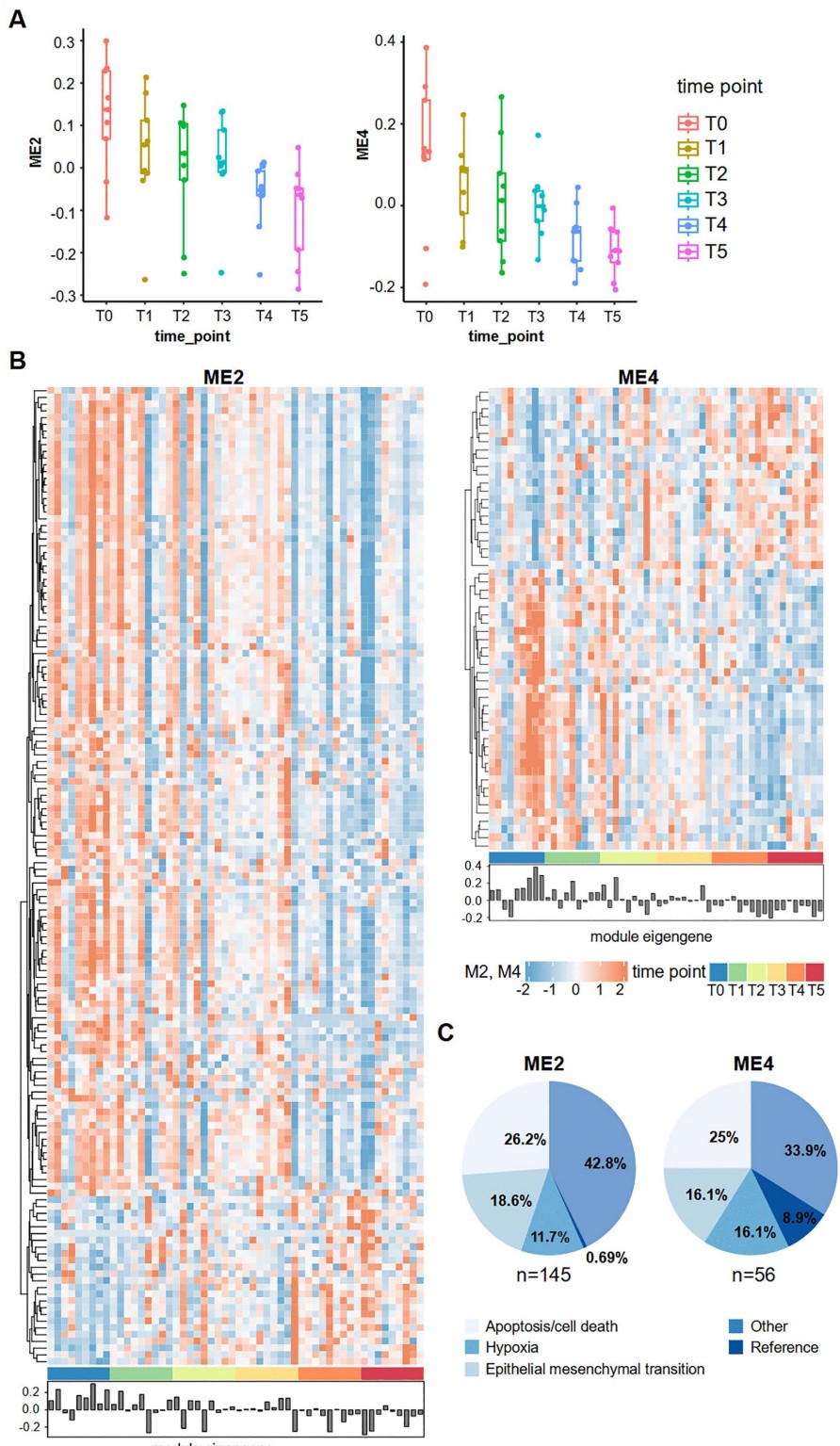

**Fig 2. Weighted Gene Co-Expression Network Analysis (WGCNA) of the expression profiles of all 54 samples using the normalized batch-corrected expression profiles of 1419 genes. A.** Two WGCNA modules significantly associated with the time course phenotype (ME2 and ME4, adjusted p-value 0.037 and 0.018, respectively) The y-axis on the boxplots represents module eigengene expression levels. **B.** Heatmaps of gene

expression of ME2 and ME4 module genes; module eigengene provide summarized representation of the expression pattern of all genes in each module and sample type. **C.** Pie charts displaying the percentage of transcripts associated with particular biological processes or functions: apoptosis/cell death, hypoxia, epithelial-mesenchymal transition, reference genes, and 'other' in each module, ME2 and ME4, respectively. T0 - time point 0, frozen immediately after resection; T1 - time point 1, frozen 10 minutes after resection; T2 - time point 2, frozen 20 minutes after resection; T3 - time point 3, frozen 30 minutes after resection; T4 - time point 4, frozen 45 minutes after resection; T5 - time point 5, frozen 60 minutes after resection.

demonstrated that CIT exceeding 15 minutes significantly impact the expression of mRNAs, proteins, and phosphosites in tumor and normal tissues across colorectal cancer, hepatocellular carcinoma, lung adenocarcinoma, and lung squamous cell carcinoma. The authors stressed that the impact of ischemia time is more pronounced at the proteomic and phosphoproteomic levels than at the mRNA level, with mRNA expression showing comparatively weaker and more homogeneous changes across cancer types [35]. Using microarrays, Musella *et al*. found minimal effects, detecting only 0.2% DEGs in normal colon tissue and colorectal carcinoma within 6 hours, though two identified DEGs were oncogenes [17]. In contrast, Spruessel *et al*. applying the same genotyping technique, reported significant changes, with 20% of detectable genes and proteins altered within 30 minutes post-excision [1]. Grizzle *et al.* noted that while most studies found only 1–3% of transcripts affected, they often overlooked the functional impact of these genes [33]. Our findings underscore that while RNA integrity remains stable within the first 30 minutes, transcriptional changes begin to emerge beyond this point. Even within a 60-minute CIT window, the occurring transcriptional changes can potentially alter the results of studies exploring cancer and adjacent tissue genetics. These observations highlight the importance of rapid tissue processing for accurate transcriptomic analyses in cancer research.

Our analysis focused on genes involved in cancer development and progression, as they are crucial for the studies of tumor margins and tissues adjacent to the tumor, where changes in the transcriptome may be subtle [21]. The results demonstrate the impact of hypoxia on EMT, a process crucial for embryonic development and tissue regeneration, which is aberrantly reactivated in cancer to drive progression and metastasis through enhanced migration, invasiveness, stemness, and resistance to therapies [36]. Genes such as *SNAI*, *CDH1*, *MMP9*, and *S100A8*, which play critical roles in the EMT in cancer, exhibited significantly altered expression levels (adjusted p-value < 0.05) in our analysis. Snail (*SNAI1*) and Slug (*SNAI2*) are transcription factors known to bind to the promoter of *CDH1*, encoding E-cadherin, to repress its transcription [37]. The suppression of *CDH1*, regulated by *SNAI*, is a key driver of EMT, contributing to cancer progression by promoting invasion and metastasis, and is recognized as a marker of malignancy and poor clinical prognosis [38]. In our study, prolonged CIT caused significant alterations in the expression of *SNAI* and *CDH1*, with upregulation of *SNAI* and downregulation of *CDH1*. Moreover, the observed upregulation of *MMP9*, likely induced by CIT, may reflect further malignant changes as MMP family members facilitate cancer cell invasion and metastasis [36]. These changes, often linked to malignancy, may be misinterpreted as early indicators of tumor progression in research on adjacent tissues. However, our findings suggest that these alterations result from prolonged CIT.

To gain an insight into complex biological mechanisms affected by CIT, we performed enrichment analysis of DEGs using two complementary approaches: a direct analysis of all DEGs and a WGCNA-based analysis of the two resulting modules, both evaluated through GO and KEGG. The direct DEG analysis provided a broad view of the pathways enriched across all DEGs, whereas WGCNA highlighted context-specific pathways within co-regulated gene modules.

GO and KEGG analyses of all DEGs identified malignancy-associated pathways, including apoptosis, ABC transporters, and TNF signaling, as well as stress response pathways (S4 Table in S1 File). In comparison, other transcriptomic studies identified alteration of pathways linked to apoptosis, stress responses, cell cycle regulation, and cancer progression [10], as well as those associated with stimulus response and signal transduction [32]. Bray *et al*. found no consistent gene ontology or pathway among 168 transcripts altered within 120 minutes of CIT, highlighting a complex cellular response [16]. In our study, despite targeting over 600 cancer-related genes, pathway enrichment was broad, supporting Bray *et al*.'s conclusions (S4 Table in S1 File).

WGCNA identified two significantly downregulated co-expressed gene modules, ME2 and ME4, with distinct functional patterns (S7 Table in S1 File). ME2 was primarily linked to apoptosis, with GO highlighting pathways like the execution phase of apoptosis and extrinsic apoptotic signaling, while KEGG linked it to FoxO, Hippo, and p53 signaling pathways. These findings underscore CIT's significant impact on apoptosis-related pathways within 60 minutes, potentially confounding studies that do not account for its effects. ME4 was enriched for metabolic pathways, including catabolic processes, homeostasis, pyruvate metabolism (GO), glycolysis/gluconeogenesis, and choline metabolism in cancer (KEGG). Both modules featured HIF-1 and cancer-related pathways, illustrating CIT's dual impact on apoptosis and metabolism, complicating transcriptomic interpretations. These findings align with reports of ischemia-induced changes in CRC, affecting oncogenes and histone-related genes involved in nucleosome organization, cell cycle, DNA replication, and p53 signaling, beginning at 8–10 minutes [1] and persisting up to 60 minutes [17]. Notably, ME4 exhibited minimal differences between T4 and T5, whereas ME2 showed a more pronounced variation between these time points, suggesting a slower reaction to CIT (Fig 2A).

Our study highlights that beyond RNA degradation, ischemic stress actively alters gene expression, emphasizing the need to standardize CIT in human tissue collection. The consistency of our observations suggests these transcriptomic changes stem from surgical and tissue handling procedures rather than malignancy. A key strength of our study is the use of NGS RNA-seq, which offers greater sensitivity and accuracy than microarrays [11,17], enabling robust detection of gene expression changes, even at low levels. Unlike previous studies, our approach analyzes over 600 genes, allowing a broader transcriptomic assessment. However, limitations include a small sample size and potential influences from enzymatic activity, warm ischemia, and other preanalytical factors [2,39]. Additionally, our analysis used a targeted cancer-related gene panel, which does not capture all transcriptomic changes.

## Conclusion

Our study highlights the significant impact of CIT on the transcriptomic profile of healthy colon mucosa, with a particular focus on cancer-related genes. Using targeted RNA sequencing and robust analyses, we identified differentially expressed genes primarily associated with apoptosis, hypoxia, and EMT, as well as pathways involved in cancer development and progression. Our findings underscore the importance of considering CIT as a critical preanalytical variable, as prolonged ischemia can alter gene expression in ways that may mimic malignant changes, leading to misinterpretations of results and incorrect clinical conclusions. Importantly, our results suggest that CIT should be kept up to 30 minutes to minimize its impact on transcriptomic profiles. However, as the effects of CIT might be tissue-dependent, this recommendation should be applied with caution when extrapolating to other tissues. While our analysis focused exclusively on transcriptomic changes, future studies employing multi-omic approaches are essential to fully characterize cold ischemia-related molecular alterations.

## Supporting information

**S1 File. Patient information, gene panel design and transcriptomic analyses results.**
(XLSX)

## Acknowledgments

We thank all the anonymous patients for acceptance to participate, sample contribution and information provided in the questionnaire. We also thank surgeons, and nurses involved in the patient recruitment process, collaborating technicians, diagnosticians and pathologists from University Clinical Centre in Gdańsk.

## Author contributions

**Conceptualization:** Anna Kostecka, Jarosław Skokowski, Leszek Kalinowski, Jan P. Dumanski, Arkadiusz Piotrowski.
**Data curation:** Natalia Filipowicz, Marcin Jąkalski.

**Formal analysis:** Katarzyna Duzowska, Marcin Jąkalski.

**Funding acquisition:** Jan P. Dumanski, Arkadiusz Piotrowski.

**Investigation:** Katarzyna Duzowska, Natalia Filipowicz, Mikołaj Opiełka, Kinga Drężek-Chyła, Monika Horbacz.

**Methodology:** Katarzyna Duzowska, Natalia Filipowicz, Mikołaj Opiełka, Anna Kostecka, Monika Horbacz, Olga Rostkowska, Jarosław Kobiela, Leszek Kalinowski, Arkadiusz Piotrowski, Marcin Jąkalski.

**Project administration:** Natalia Filipowicz, Jan P. Dumanski, Arkadiusz Piotrowski.

**Resources:** Kinga Drężek-Chyła, Jarosław Skokowski, Olga Rostkowska, Jarosław Kobiela.

**Software:** Marcin Jąkalski.

**Supervision:** Natalia Filipowicz, Jan P. Dumanski, Arkadiusz Piotrowski.

**Visualization:** Natalia Filipowicz, Marcin Jąkalski.

**Writing – original draft:** Katarzyna Duzowska, Natalia Filipowicz, Arkadiusz Piotrowski, Marcin Jąkalski.

**Writing – review & editing:** Katarzyna Duzowska, Natalia Filipowicz, Mikołaj Opiełka, Kinga Drężek-Chyła, Anna Kostecka, Monika Horbacz, Jarosław Skokowski, Olga Rostkowska, Jarosław Kobiela, Leszek Kalinowski, Jan P. Dumanski, Arkadiusz Piotrowski, Marcin Jąkalski.

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
