## [Decision Letter · Decision Letter 0]

17 Oct 2025

PONE-D-25-11860

The effect of cold ischemia time on hypoxia, EMT, and apoptosis pathways in normal colon mucosa

PLOS ONE

Dear Dr. Filipowicz,

Thank you for submitting your manuscript to PLOS ONE. After careful consideration, we have decided that your manuscript does not meet our criteria for publication and must therefore be rejected.

Specifically:

After careful consideration of the reviewer comments and editorial assessment, we regret to inform you that we are unable to proceed with publication.

Your findings on how cold ischemia time (CIT) may affect gene expression in normal colon mucosa are clearly presented and technically sound. However, the study remains largely descriptive, and the conclusions regarding cancer-related transcriptional changes are not directly supported by functional validation. Moreover, the focus on normal tissue limits the broader applicability of the findings.

Given these limitations, we do not believe the manuscript meets the publication criteria for PLOS ONE in its current form. We thank you again for considering the journal and encourage you to further develop this work for submission elsewhere.

I am sorry that we cannot be more positive on this occasion, but hope that you appreciate the reasons for this decision.

Kind regards,

Yingkun Xu, M.D

Academic Editor

PLOS ONE

Additional Editor Comments (if provided):

Thank you for submitting your manuscript to PLOS ONE. We appreciate the thoughtful study design and attention to data quality.

After careful consideration of the reviewer comments and editorial assessment, we regret to inform you that we are unable to proceed with publication.

Your findings on how cold ischemia time (CIT) may affect gene expression in normal colon mucosa are clearly presented and technically sound. However, the study remains largely descriptive, and the conclusions regarding cancer-related transcriptional changes are not directly supported by functional validation. Moreover, the focus on normal tissue limits the broader applicability of the findings.

Given these limitations, we do not believe the manuscript meets the publication criteria for PLOS ONE in its current form. We thank you again for considering the journal and encourage you to further develop this work for submission elsewhere.

Reviewers' comments:

Reviewer's Responses to Questions

**Comments to the Author**

1. Is the manuscript technically sound, and do the data support the conclusions?

Reviewer #1: Yes

Reviewer #2: Yes

2. Has the statistical analysis been performed appropriately and rigorously?

Reviewer #1: Yes

Reviewer #2: Yes

3. Have the authors made all data underlying the findings in their manuscript fully available?

Reviewer #1: Yes

Reviewer #2: Yes

4. Is the manuscript presented in an intelligible fashion and written in standard English?

Reviewer #1: Yes

Reviewer #2: Yes

Reviewer #1: accepted.

explain your answers to the questions above. You may also include additional comments for the author, including concerns about dual publication, research ethics, or publication ethics. (Please upload your review as an attachment if it exceeds 20,000 characters) (Limit 200 to 20000 Characters)

Reviewer #2: The paper explains the effect of cold ischemia time (CIT) and its importance in transcriptomic studies and reproducibility. The study objective evaluating CIT effects on cancer-related genes is relevant and practical. The findings from this study suggest a CIT limit (≤45–60 min), information that is valuable for the research community.

The number of samples, patient count and CIT intervals are clearly described but whether all 54 samples were paired per patient across all time points needs further clarification.

Overall, the analysis is comprehensive. The combination of RNA-seq, functional enrichment and WGCNA strengthens the biological interpretation. Identifying DEGs and linking them to apoptosis, hypoxia, EMT and cancer-related pathways is relevant and informative.

**Do you want your identity to be public for this peer review?** For information about this choice, including consent withdrawal, please see our Privacy Policy

Reviewer #1: No

Reviewer #2: No

- - - - -

---

## [Author Response · Author response to Decision Letter 1]

3 Nov 2025

1. Is the manuscript technically sound, and does the data support the conclusions?

Reviewer #1: Yes

Reviewer #2: Yes

2. Has the statistical analysis been performed appropriately and rigorously?

Reviewer #1: Yes

Reviewer #2: Yes

3. Have the authors made all data underlying the findings in their manuscript fully available?

Reviewer #1: Yes

Reviewer #2: Yes

4. Is the manuscript presented in an intelligible fashion and written in standard English?

Reviewer #1: Yes

Reviewer #2: Yes

5. Review Comments to the Author

Reviewer #1: accepted.

explain your answers to the questions above. You may also include additional comments for the author, including concerns about dual publication, research ethics, or publication ethics. (Please upload your review as an attachment if it exceeds 20,000 characters) (Limit 200 to 20000 Characters)

Reviewer #2: The paper explains the effect of cold ischemia time (CIT) and its importance in transcriptomic studies and reproducibility. The study objective evaluating CIT effects on cancer-related genes is relevant and practical. The findings from this study suggest a CIT limit (≤45–60 min), information that is valuable for the research community.

The number of samples, patient count and CIT intervals are clearly described but whether all 54 samples were paired per patient across all time points needs further clarification.

Answer: All 54 samples used in our study originate from 9 CRC patients, with samples collected at 6 distinct time points as defined in the manuscript (9 patients × 6 time points = 54 samples). Thus, each patient contributed one sample per time point, resulting in a fully paired dataset.

We accounted for the paired nature of the RNAseq data by treating the patient ID as batch and removing its effect from the data with ComBat-seq. The sentence clarifying this was modified in lines 173-175.

Overall, the analysis is comprehensive. The combination of RNA-seq, functional enrichment and WGCNA strengthens the biological interpretation. Identifying DEGs and linking them to apoptosis, hypoxia, EMT and cancer-related pathways is relevant and informative.

6. PLOS authors have the option to publish the peer review history of their article (what does this mean?). If published, this will include your full peer review and any attached files.

Do you want your identity to be public for this peer review? For information about this choice, including consent withdrawal, please see our Privacy Policy.

Reviewer #1: No

Reviewer #2: No

---

## [Decision Letter · Decision Letter 1]

30 Dec 2025

Dear Dr. Filipowicz,

Thank you for submitting your manuscript to PLOS ONE. After careful consideration, we feel that it has merit but does not fully meet PLOS ONE’s publication criteria as it currently stands. Therefore, we invite you to submit a revised version of the manuscript that addresses the points raised during the review process.

As you can see from the reviews, two of the reviewers have recommended acceptance while a third one has raised some concerns. Therefore, we are asking you to address those concerns in a revised version. If some or all of these concerns cannot be addressed, please provide clear justification about why that is not possible.

plosone@plos.org . A letter that responds to each point raised by the academic editor and reviewer(s). You should upload this letter as a separate file labeled 'Response to Reviewers'.A marked-up copy of your manuscript that highlights changes made to the original version. You should upload this as a separate file labeled 'Revised Manuscript with Track Changes'.An unmarked version of your revised paper without tracked changes. You should upload this as a separate file labeled 'Manuscript'.

We look forward to receiving your revised manuscript.

Kind regards,

Aniruddha Datta

Academic Editor

PLOS One

Journal Requirements:

"This study was supported by the Foundation for Polish Science (FNP) under the International Research Agendas Program to J.P.D. and A.P., co-financed by the European Union under the European Regional Development Fund (MAB/2018/6); and the Ministry of Science and Higher Education, (grant No. 2/566516/ SPUB/SP/2023) to L.K.

4. We note that there is identifying data in the supporting information <Duzowska_et_al_Supplementary_tables.xlsx>. Due to the inclusion of these potentially identifying data, we have removed this file from your file inventory. Prior to sharing human research participant data, authors should consult with an ethics committee to ensure data are shared in accordance with participant consent and all applicable local laws.

-Location data

Please remove or anonymize all personal information, ensure that the data shared are in accordance with participant consent, and re-upload a fully anonymized data set. Please note that spreadsheet columns with personal information must be removed and not hidden as all hidden columns will appear in the published file.

Additional Editor Comments (if provided):

Reviewers' comments:

Reviewer's Responses to Questions

**Comments to the Author**

Reviewer #3: All comments have been addressed

Reviewer #4: (No Response)

Reviewer #5: All comments have been addressed

2. Is the manuscript technically sound, and do the data support the conclusions?

Reviewer #3: Yes

Reviewer #4: Yes

Reviewer #5: Yes

3. Has the statistical analysis been performed appropriately and rigorously?

Reviewer #3: Yes

Reviewer #4: Yes

Reviewer #5: Yes

4. Have the authors made all data underlying the findings in their manuscript fully available?

Reviewer #3: Yes

Reviewer #4: Yes

Reviewer #5: Yes

5. Is the manuscript presented in an intelligible fashion and written in standard English?

Reviewer #3: Yes

Reviewer #4: Yes

Reviewer #5: Yes

Reviewer #3: The authors have updated their manuscript with the comments and questions raised by the reviewers. I have no further feedback.

Reviewer #4: Summary

This study investigates the effect of cold ischemia time (CIT) on the transcriptomic profile of normal colonic mucosa, with specific attention to cancer-related pathways such as apoptosis, hypoxia, and epithelial–mesenchymal transition (EMT). The authors analyzed 54 normal mucosal samples from 9 colorectal cancer patients and performed RNA-seq under CIT conditions ranging from 0 to 60 minutes. The manuscript describes the experimental workflow, RNA quality assessment, differential expression analysis, WGCNA, and pathway enrichment.

In the Results, the authors report 44 significant DEGs between T0 and T5, enriched in apoptosis, hypoxia, and EMT pathways. They note minimal transcriptomic changes before T3, a small increase at T4 (6 DEGs), and a larger increase at T5. The Discussion proposes that CIT can induce tumor-like molecular signals and stresses the importance of controlling ischemia time during tissue acquisition. The authors recommend keeping CIT under 45 minutes.

Questions and Concerns

The proposed “safe CIT threshold” appears overly optimistic and contradicts multiple prior studies.

The manuscript recommends limiting CIT to 45–60 minutes to minimize molecular artifacts. However, previous literature shows that both gene and protein expression in colonic tissue can be significantly altered within minutes.

• Spruessel et al. demonstrated that a few minutes to 30 minutes of ischemia can markedly change gene and protein expression in colon samples:

https://pubmed.ncbi.nlm.nih.gov/15211754/

• von der Heyde et al. (2024, Cell Death & Disease, Nature Publishing Group) showed that proteomic dysregulation emerges above 15 minutes of ischemia:

“Specimen ischemia time above 15 min is mostly associated with a dysregulation of proteins in the immune-response pathway.”

https://www.nature.com/articles/s41419-024-07090-x

These findings suggest that the authors’ proposed CIT tolerance window may not be sufficiently conservative.

The study relies solely on mRNA measurements and lacks multi-omic validation.

No proteomic, immunohistochemical, or additional orthogonal assays are provided. This is a notable gap because proteomic alterations often occur earlier and more sensitively than transcriptomic changes.

For example, von der Heyde et al. (2024) report:

“The proteomics analysis revealed that specimen ischemia time above 15 min is mostly associated with a dysregulation of proteins in the immune-response pathway.”

https://www.nature.com/articles/s41419-024-07090-x

Without multi-omics confirmation, it is difficult to determine whether the reported mRNA changes reflect true biological responses or early degradation phenomena.

Warm ischemia time (WIT) is acknowledged but not recorded or modeled, which is inconsistent with established standards.

Although the authors state that WIT is important, they do not:

• document each patient’s WIT;

• include WIT as a covariate in statistical models (e.g., edgeR, linear models);

• analyze interactions between WIT and CIT.

This is problematic because prior work has emphasized that WIT must be explicitly accounted for. Musella et al. (2013) show that both WIT and CIT shape molecular stability in colon tissue:

“A critical time point for tissue handling in colon seems to be 60 minutes at room temperature … accounting for warm ischemia in this tumor type.”

https://journals.plos.org/plosone/article?id=10.1371/journal.pone.0053406

Without controlling for WIT, the attribution of transcriptional changes solely to CIT is not well supported.

Reviewer #5: This manuscript presents a well-designed and carefully executed study examining the impact of cold ischemia time (CIT) on transcriptomic profiles of normal colon mucosa. The topic is timely and highly relevant, as preanalytical variables such as CIT remain a major source of variability and irreproducibility in transcriptomic and biobanking-based research.

The experimental design is a clear strength of the study. The use of paired samples collected from the same patients at multiple predefined time points provides strong internal control and allows for a robust assessment of time-dependent transcriptional changes. The application of RNA sequencing combined with functional enrichment analysis and WGCNA is appropriate and well justified, and the statistical analyses are performed rigorously and transparently. The clarification regarding the fully paired nature of the dataset and the use of ComBat-seq to account for patient-specific effects adequately addresses potential concerns about confounding.

The results are clearly presented and convincingly demonstrate that prolonged CIT induces transcriptional changes in pathways related to hypoxia, apoptosis, EMT, and cancer-associated signaling, even in morphologically normal tissue. The identification of a practical CIT threshold (≤45–60 minutes) is particularly valuable and provides actionable guidance for researchers involved in tissue collection, biobanking, and transcriptomic analyses.

The manuscript is generally well written, logically structured, and accessible to a broad readership. The discussion appropriately contextualizes the findings within existing literature and highlights both the strengths and limitations of the study, including sample size and the use of a targeted gene panel.

Overall, this work represents a solid and meaningful contribution to the field of preanalytical variability and transcriptomic research. The conclusions are well supported by the data, and the study should be of broad interest to researchers working with human tissues, particularly in cancer and biobanking contexts.

**Do you want your identity to be public for this peer review?** For information about this choice, including consent withdrawal, please see our Privacy Policy

Reviewer #3: No

Reviewer #4: No

Reviewer #5: No

---

## [Author Response · Author response to Decision Letter 2]

26 Jan 2026

PONE-D-25-11860R1 “The effect of cold ischemia time on hypoxia, EMT, and apoptosis pathways in normal colon mucosa”

Answers to the comments:

Comment 1:

The proposed “safe CIT threshold” appears overly optimistic and contradicts multiple prior studies. The manuscript recommends limiting CIT to 45–60 minutes to minimize molecular artifacts. However, previous literature shows that both gene and protein expression in colonic tissue can be significantly altered within minutes.

• Spruessel et al. demonstrated that a few minutes to 30 minutes of ischemia can markedly change gene and protein expression in colon samples: https://pubmed.ncbi.nlm.nih.gov/15211754/

• von der Heyde et al. (2024, Cell Death & Disease, Nature Publishing Group) showed that proteomic dysregulation emerges above 15 minutes of ischemia:

“Specimen ischemia time above 15 min is mostly associated with a dysregulation of proteins in the immune-response pathway.”

https://www.nature.com/articles/s41419-024-07090-x

These findings suggest that the authors’ proposed CIT tolerance window may not be sufficiently conservative.

Answer 1:

Thank you for this valuable comment.

Our original recommendation of a 45–60 minutes CIT threshold was based strictly on the results obtained in the present study, which employed a targeted transcriptomic approach covering 634 genes primarily involved in apoptosis, cell death, hypoxia, epithelial to mesenchymal transition (EMT), and cancer related pathways. Within this restricted gene set and thorough analytical framework, we did not observe statistically significant transcriptional changes within the first 30 minutes post excision. The earliest detectable alterations were observed between 30 and 45 minutes, with a substantially larger number of differentially expressed genes emerging after 45 minutes.

After careful consideration of the reviewer’s comment and the cited literature, we agree that a more conservative recommendation is warranted. We have therefore revised the manuscript to propose a stricter CIT threshold of ≤30 minutes, which better aligns with existing evidence and reflects a precautionary approach. Corresponding changes have been made in the manuscript (lines 50 and 402).

We also acknowledge that our findings may appear contradictory to several previous studies on CIT. This discrepancy is already discussed in the Discussion section, where we cite selection of articles on CIT (lines 304-334) including Spruessel et al. and von der Heyde et al. (2024). We now modified slightly this section by adding information from von der Heyde et al. on ischemia time being more pronounced at the proteomic and phosphoproteomic levels than at the mRNA level (lines 319-321). We would also like to stress that we do not claim in our paper that the CIT threshold guidelines would be applicable to all kinds of studies, including proteomics.

We believe that the revised recommendation of a ≤30 minute CIT threshold, along with a clearer contextualization of our results to other transcriptomic and proteomic studies, strengthens the manuscript.

Comment 2:

The study relies solely on mRNA measurements and lacks multi-omic validation. No proteomic, immunohistochemical, or additional orthogonal assays are provided. This is a notable gap because proteomic alterations often occur earlier and more sensitively than transcriptomic changes.

For example, von der Heyde et al. (2024) report: “The proteomics analysis revealed that specimen ischemia time above 15 min is mostly associated with a dysregulation of proteins in the immune-response pathway. ”https://www.nature.com/articles/s41419-024-07090-x

Without multi-omics confirmation, it is difficult to determine whether the reported mRNA changes reflect true biological responses or early degradation phenomena.

Answer 2:

We appreciate this thoughtful comment highlighting the importance of multi-omic approaches in assessing tissue quality and preanalytical variables. We agree that proteomic and other assays can offer complementary insights, particularly regarding early ischemia-related changes that may not be immediately reflected at the transcript level. However, our study was intentionally focused on transcriptomic profiling, using a targeted panel of 634 genes related to apoptosis, hypoxia, cell death, and cancer progression. This approach, although narrower than full transcriptome or proteome analysis, was chosen for its clinical relevance and feasibility in standardizing preanalytical workflows. Due to technical and resource constraints, multi-omic validation was beyond the scope of the current study. We believe that our results offer a practical, transcriptome-focused reference point that can guide tissue handling protocols and inform future integrative studies. To stress limitation of our approach we added additional sentence in the Conclusion section (lines 405-407).

Comment 3:

Warm ischemia time (WIT) is acknowledged but not recorded or modeled, which is inconsistent with established standards. Although the authors state that WIT is important, they do not:

• document each patient’s WIT;

• include WIT as a covariate in statistical models (e.g., edgeR, linear models);

• analyze interactions between WIT and CIT.

This is problematic because prior work has emphasized that WIT must be explicitly accounted for. Musella et al. (2013) show that both WIT and CIT shape molecular stability in colon tissue: “A critical time point for tissue handling in colon seems to be 60 minutes at room temperature … accounting for warm ischemia in this tumor type.”

https://journals.plos.org/plosone/article?id=10.1371/journal.pone.0053406

Without controlling for WIT, the attribution of transcriptional changes solely to CIT is not well supported.

Answer 3:

Thank you very much for this important comment. We fully agree that warm ischemia time (WIT) is a critical preanalytical factor and appreciate the opportunity to clarify our rationale and study limitations regarding this aspect.

As noted in the introduction (lines 56–70), we acknowledge the impact of WIT on gene expression and reference prior studies addressing its role. In our setting, WIT could not be precisely measured for each patient due to the clinical priority of ensuring a safe and effective surgical outcome. WIT, defined as the period during which the tissue remains in the body without effective oxygen supply, is inherently difficult to standardize and control, especially during open surgical procedures.

In case of our cohort all the procedures were performed using an open surgical approach, which minimizes warm ischemia time (WIT) by allowing immediate specimen extraction following devascularization. In contrast, laparoscopic and robotic approaches require placement of the devascularized specimen into an extraction bag, with removal occurring only at the completion of the procedure. In the present case series, all operations consisted of left-sided colectomies or rectal resections, with operative times ranging from 1 hour 15 minutes to 3 hours 20 minutes. Considering the open approach and the review specific technical aspects of the procedures, it is reasonable to assume that WIT did not exceed 40 minutes in any case. This variable should be considered unmodifiable, as it is largely determined by intra-abdominal anatomical conditions.

We respectfully note that although we cited Musella et al. (2013) to highlight preanalytical variability, the ischemia time discussed in their study appears to align more closely with cold ischemia time (CIT), not WIT. In the methods section, they state that “the time was measured starting from patient’s surgical excision and the first timepoint analysed (T0)”, indicating that their 60-minute measurement refers to ex-vivo conditions.

Taking all of these into account we recognized the importance of addressing this limitation in a transparent way. We have now added a suitable statement regarding WIT in the Materials and Methods: “Study Design and Sample Handling” (lines 116-124) emphasizing the use of open surgery, estimated upper limit of WIT and difficulty recording patient-specific WIT under current surgical protocols.

We hope this provides clarity and adequately addresses the reviewer’s concern.

---

## [Decision Letter · Decision Letter 2]

4 Mar 2026

The effect of cold ischemia time on hypoxia, EMT, and apoptosis pathways in normal colon mucosa

PONE-D-25-11860R2

Dear Dr. Filipowicz,

We’re pleased to inform you that your manuscript has been judged scientifically suitable for publication and will be formally accepted for publication once it meets all outstanding technical requirements.

Kind regards,

Aniruddha Datta

Academic Editor

PLOS One

Additional Editor Comments (optional):

Reviewers' comments:

Reviewer's Responses to Questions

**Comments to the Author**

Reviewer #4: All comments have been addressed

2. Is the manuscript technically sound, and do the data support the conclusions?

Reviewer #4: Yes

3. Has the statistical analysis been performed appropriately and rigorously?

Reviewer #4: Yes

4. Have the authors made all data underlying the findings in their manuscript fully available?

Reviewer #4: Yes

5. Is the manuscript presented in an intelligible fashion and written in standard English?

Reviewer #4: Yes

Reviewer #4: The authors have adequately addressed the major concerns. The recommended CIT threshold has been revised to ≤30 minutes and appropriately contextualized relative to prior literature. The conclusions are now clearly restricted to transcriptomic findings within the targeted gene panel.

Limitations regarding the lack of multi-omic validation and the absence of patient-specific WIT measurements are transparently acknowledged. While these constraints remain, they do not undermine the validity of the presented data within the defined scope.

**Do you want your identity to be public for this peer review?** For information about this choice, including consent withdrawal, please see our Privacy Policy

Reviewer #4: No

---

## [Editor Report · Acceptance letter]

PONE-D-25-11860R2

PLOS One

Dear Dr. Filipowicz,

I'm pleased to inform you that your manuscript has been deemed suitable for publication in PLOS One. Congratulations! Your manuscript is now being handed over to our production team.

Kind regards,

on behalf of

Dr. Aniruddha Datta

Academic Editor

PLOS One